# Expanding Holographic Embeddings for Knowledge Completion

**Yexiang Xue**[⋆]     **Yang Yuan**[†]     **Zhitian Xu**[⋆]     **Ashish Sabharwal**[‡]

[⋆] Dept. of Computer Science, Purdue University, West Lafayette, IN, USA
[†] Dept. of Computer Science, Cornell University, Ithaca, NY, USA
[‡] Allen Institute for Artificial Intelligence (AI2), Seattle, WA, USA

## Abstract

Neural models operating over structured spaces such as knowledge graphs require a continuous embedding of the discrete elements of this space (such as entities) as well as the relationships between them. Relational embeddings with high expressivity, however, have high model complexity, making them computationally difficult to train. We propose a new family of embeddings for knowledge graphs that interpolate between a method with high model complexity and one, namely Holographic embeddings (HOLE), with low dimensionality and high training efficiency. This interpolation, termed HOLEX, is achieved by concatenating several linearly perturbed copies of original HOLE. We formally characterize the number of perturbed copies needed to provably recover the full entity-entity or entity-relation interaction matrix, leveraging ideas from Haar wavelets and compressed sensing. In practice, using just a handful of Haar-based or random perturbation vectors results in a much stronger knowledge completion system. On the Freebase FB15K dataset, HOLEX outperforms originally reported HOLE by 14.7% on the HITS@10 metric, and the current path-based state-of-the-art method, PTransE, by 4% (absolute).

## 1   Introduction

Relations, as a key concept in artificial intelligence and machine learning, allow human beings as well as intelligent systems to learn and reason about the world. In particular, relations among multiple entities and concepts enable us to make logical inference, learn new concepts, draw analogies, make comparisons, etc. This paper considers relational learning for knowledge graphs (KGs), which often contain knowledge in the form of binary relations, such as *livesIn(Bill Gates, Seattle)*. A number of very large KGs, with millions and even billions of facts, have become prominent in the last decade, such as Freebase [3], DBpedia [2], YAGO [11], WordNet [17], and WebChild [26].

KGs can be represented as a multigraph, where entities such as *Bill Gates* and *Seattle* are nodes, connected with zero or more relations such as *livesIn* and *likes*. Facts such as *livesIn(Bill Gates, Seattle)* form typed edges, with the relation—in this case *livesIn*—being the edge type. In particular, we are interested in the *knowledge completion* task for KGs: Given an existing KG, we would like to use statistical machine learning tools to extract correlations among its entities and relations, and use these correlations to derive new knowledge about them.

Compositional vector space models, also referred to as matrix or tensor factorization based methods, have proven to be highly effective for KG completion [e.g., 4, 5, 7, 8, 12, 14–16, 18, 19, 23–25, 28]. In these models, entities and relations are represented as (learned) vectors in a high dimensional space, and various forms of compositional operators are used to determine the likelihood of a candidate fact. A good design of the compositional operator is often key to the success of the model. Such design

must balance computational complexity against model complexity. Not surprisingly, embedding models capable of capturing rich correlations in relational data often have limited computational scalability. On the other hand, models that can be trained efficiently are often less expressive.

We focus on two compositional operators. The first is the *full tensor product* [18], which captures correlations between every pair of dimensions of two embedding vectors in $\mathbb{R}^d$, by considering their outer product. The resulting quadratic ($d^2$) parameter space makes this impractical even for modestly sized KGs. The second is the circular correlation underlying *holographic embedding* or HOLE [19], which is inspired by holographic models of associated memory. Notably, HOLE keeps the parameter space linear in $d$ by capturing only the sum along each diagonal of the full tensor product matrix.

Our main contribution is a new compositional operator that combines the strengths of these two models, resulting in much stronger knowledge completion system. Specifically, we propose *expanded holographic embeddings* or HOLEX, which is a collection of models that interpolates between holographic embeddings and the full tensor product.

The idea is to concatenate $l \geq 1$ copies of the HOLE model, each perturbed by a linear vector, allowing various copies to focus on different subspaces of the embedding. HOLEX forms a complete spectrum connecting HOLE with the full tensor product model: it falls back to HOLE when $l = 1$ and all entries in the perturbation vector are non-zero, and is equivalent to the full tensor product model when $l = d$, the embedding dimension, and all perturbation vectors are linearly independent.

We consider two families of perturbation vectors, low frequency Haar wavelets [6, 10] and random 0/1 vectors. We show that using the former corresponds to considering sums of multiple subsequences of each diagonal line of the full product matrix, in contrast to the original holographic embedding, which sums up the entire diagonal. We find that even just a few low frequency vectors in the Haar matrix are quite effective in practice for HOLEX. When using the complete Haar matrix, the length of the subsequences becomes one, thereby recovering the full tensor product case. Our second family of perturbation vectors, namely random 0/1 vectors, corresponds to randomly sub-selecting half the rows of the tensor product matrix in each copy. This is valuable when the full product matrix is sparse. Specifically, using techniques from compressed sensing, if each diagonal line is dominated by a few large entries (in terms of absolute values), we show that a logarithmic number of random vectors suffice to recover information from these large entries.

To assess its efficacy, we implement HOLEX using the framework of ProjE [23], a recent neural method developed for the Freebase FB15K knowledge completion dataset [3, 5], where the 95% confidence interval for statistical significance is 0.3%. In terms of the standard HITS@10 metric, HOLEX using 16 random 0/1 vectors outperforms the original HOLE by 14.7% (absolute), ProjE by 5.7%, and a path-based state-of-the-art method by 4%.

## 2    Preliminaries

We use knowledge graphs to predict new relations between entities. For example, given entities *Albany* and the *New York State*, possible relationships between these two entities are *CityIn* and *CapitalOf*. Formally, let $\mathcal{E}$ denote the set of all entities in a KG $\mathcal{G}$. A relation $r$ is a subset of $\mathcal{E} \times \mathcal{E}$, corresponding to all entity pairs that satisfy the relation. For example, the relation *CapitalOf* contains all (*City*, *State*) pairs in which the *City* is the capital of that particular *State*. For each relation $r$, we would like to learn the characterization function for $r$, $\phi_r(s, o)$, which evaluates to +1 if the entity pair $(s, o)$ is in the relation set, otherwise, -1. Notice that $s$ and $o$ are typically asymmetrical. For example, Albany is the capital of the New York State, but not the other way around. Relations can be visualized as a knowledge graph, where the nodes represent entities, and one relation corresponds to a set of edges connecting entity pairs with the given relation.

As mentioned earlier, compositional embeddings are useful models for prediction in knowledge graphs. Generally speaking, these models embed entities as well as relations jointly into a high dimensional space. Let $s \in \mathbb{R}^{d_s}$, $o \in \mathbb{R}^{d_o}$, $r \in \mathbb{R}^{d_r}$ be the embeddings for entities $s$ and $o$, and the relation $r$, respectively. Compositional embeddings learn a score function $\sigma(.)$ that approximates the posterior probability of $\phi_r(s, o)$ conditioned on the dataset $\Omega$:

$$\Pr\left(\phi_r(s, o) = 1 \mid \Omega\right) = \sigma(\boldsymbol{s}, \boldsymbol{o}, \boldsymbol{r}). \tag{1}$$

Many models have been proposed with different functional forms for $\sigma$ [e.g., 4, 5, 8, 12, 15, 16, 18, 19, 23–25, 28]. A crucial part of these models is the *compositional operators* they use to

capture the correlation between entities and relations. Given entities (and/or relations) embeddings $\boldsymbol{a} = (a_0, \ldots, a_{d_a-1})' \in \mathbb{R}^{d_a}$ and $\boldsymbol{b} = (b_0, \ldots, b_{d_b-1})' \in \mathbb{R}^{d_b}$, a *compositional operator* is a function $f : \mathbb{R}^{d_a} \times \mathbb{R}^{d_b} \to \mathbb{R}^{d_f}$, which maps $\boldsymbol{a}$ and $\boldsymbol{b}$ into another high dimensional space[1]. Such operators are used to combine the information from the embeddings of entities and relations to predict the likelihood of a particular entity-relation tuple in the score function. A good compositional operator not only extracts information effectively from $\boldsymbol{a}$ and $\boldsymbol{b}$, but also trades it off with model complexity.

One approach is to use **vector arithmetic operations**, such as (weighted) vector addition and subtraction used by TransE [5], TransH [28], and ProjE [23]. One drawback of this approach is that the embedding dimensions remain independent in such vector operations, preventing the model from capturing rich correlations across different dimensions. Another popular compositional operator is to **concatenate** the embeddings of relations and entities, and later apply a non-linear activation function to implicitly capture correlations [8, 24].

Given the importance of capturing rich correlations, we focus on two representative compositional operators that explicitly model the correlations among entities and relations: the full tensor product and the holographic embedding, described below.

**Full Tensor Product**  Many models, such as RESCAL [18] and its compositional training extension [9] and Neural Tensor Network [25], take the full tensor product as the compositional operator. Given two embedding vectors $\boldsymbol{a}, \boldsymbol{b} \in \mathbb{R}^d$, the full tensor product is defined as $\boldsymbol{a} \otimes \boldsymbol{b} = \boldsymbol{a}\boldsymbol{b}^T$, i.e.,

$$[\boldsymbol{a} \otimes \boldsymbol{b}]_{i,j} = a_i b_j. \tag{2}$$

The full tensor product captures all pairwise multiplicative interactions between $\boldsymbol{a}$ and $\boldsymbol{b}$. Intuitively, a feature in $\boldsymbol{a} \otimes \boldsymbol{b}$ is "on" (with large absolute value), if and only if the corresponding features in *both* $\boldsymbol{a}$ and $\boldsymbol{b}$ are "on". This helps entities with multiple characteristics. For example, consider an entity *Obama*, who is a man, a basketball player, and a former president of the US. In the embeddings for *Obama*, we can have one dimension firing up when it is coupled with *Chicago Bulls* (basketball team), but a different dimension firing up when coupled with the *White House*.

However, this rich expressive power comes at a cost: a huge parameter space, which makes it difficult, if not impossible, to effectively train a model on large datasets. For example, for RESCAL, the score for a triple $(s, r, o)$ is defined as:

$$\sigma(\boldsymbol{s}, \boldsymbol{o}, \boldsymbol{r}) = \mathrm{grandsum}\left((\boldsymbol{a} \otimes \boldsymbol{b}) \circ W_r\right) \tag{3}$$

where $W_r \in \mathbb{R}^{d \times d}$ is the matrix encoding for relation $r$, $\circ$ refers to the Hadamard product (i.e., the element-wise product), and $\mathrm{grandsum}$ refers to the sum of all entries of a matrix. With $|R|$ relations, the number of parameters is dominated by the embedding of all relations, totaling $d^2|R|$. This quickly becomes infeasible even for modestly sized knowledge graphs.

**Holographic Embedding**  HOLE provides an alternative compositional operator using the idea of *circular correlation*. Given $\boldsymbol{a}, \boldsymbol{b} \in \mathbb{R}^d$, the holographic compositional operator $h : \mathbb{R}^d \times \mathbb{R}^d \to \mathbb{R}^d$ produces an interaction vector of the same dimension as $\boldsymbol{a}$ and $\boldsymbol{b}$, with the $k$-th dimension being:

$$h_k(\boldsymbol{a}, \boldsymbol{b}) = [\boldsymbol{a} \star \boldsymbol{b}]_k = \sum_{i=0}^{d-1} a_i b_{(i+k) \bmod d}. \tag{4}$$

Figure 1 (left) provides a graphical illustration. HOLE computes the sum of each (circular) diagonal line of the original tensor product matrix, collapsing a two-dimensional matrix into a one-dimensional vector. Put another way, HOLE still captures pairwise interactions between different dimensions of $\boldsymbol{a}$ and $\boldsymbol{b}$, but collapses everything along a each individual diagonal and retains only the sum for each such 'bucket'. The HOLE score for a triple $(s, r, o)$ is defined as:

$$\sigma(\boldsymbol{s}, \boldsymbol{o}, \boldsymbol{r}) = (\boldsymbol{s} \star \boldsymbol{o}) \cdot \boldsymbol{r} \tag{5}$$

where $\cdot$ denotes dot-product. This, requires only $d|R|$ parameters for encoding all relations.[2]

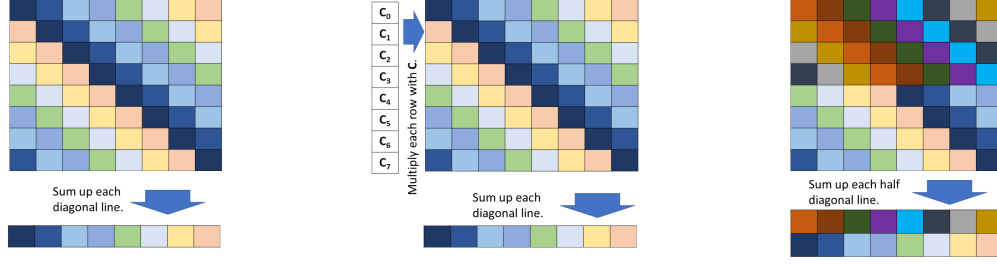

Figure 1: (Left) Visualization of HOLE, which collapses the full tensor product $M = \boldsymbol{a}\boldsymbol{b}'$ into a vector by summing up along each (circular) diagonal line, depicted with the same color. (Middle) HOLE perturbed with a vector $c$, where each row of $M$ is multiplied with one entry in $c$ prior to the holographic operation. (Right) HOLEX using the first two Haar vectors. When $M$ has dimension $d \times d$, this is equivalent to returning a $2 \times d$ matrix that sums up along each half of each diagonal line, depicted by the same color.

The circular correlation used in Holographic embedding can be seen as a projection of the full tensor product by weighting all interactions the same along each diagonal line. Given its similarity to (circular) convolution, the actual computation can be carried out efficiently with the fast Fourier transformation (FFT): $h(\boldsymbol{a}, \boldsymbol{b}) = \mathcal{F}^{-1}\left(\overline{\mathcal{F}(\boldsymbol{a})} \circ \mathcal{F}(\boldsymbol{b})\right)$. As before, $\circ$ refers to element-wise product. $\mathcal{F}$ is the discrete Fourier transform. $\overline{(x)}$ represents the complex conjugate of $x$.

## 3 Expanding Holographic Embeddings

Is there a model that sits in between HOLE and the full tensor product, and *provides a better trade-off than either extreme between computational complexity and model complexity*? We present Expanded Holographic Embeddings or HOLEX, which is a collection of models with increasing complexity that provides a controlled way to interpolate HOLE and the full tensor product. Given a fixed vector $\boldsymbol{c} \in \mathbb{R}^d$, we define the *perturbed holographic compositional operator* for $\boldsymbol{a}, \boldsymbol{b} \in \mathbb{R}^d$ as:

$$h(\boldsymbol{a}, \boldsymbol{b}; \boldsymbol{c}) = (\boldsymbol{c} \circ \boldsymbol{a}) \star \boldsymbol{b}. \tag{6}$$

As before, $\circ$ represents the Hadamard product and the score for a triple $(s, r, o)$ is computed by taking the dot product of this composition between two elements (e.g., $s$ and $o$) and a $d$-dimensional vector encoding the third element (e.g., $r$). In other words, the $k$-th dimension of $h$ now becomes:

$$h_k(\boldsymbol{a}, \boldsymbol{b}; \boldsymbol{c}) = [(\boldsymbol{c} \circ \boldsymbol{a}) \star \boldsymbol{b}]_k = \sum_{i=0}^{d-1} c_i a_i b_{(i+k) \bmod d}. \tag{7}$$

In practice, vector $\boldsymbol{c}$ is chosen prior to training. As depicted in Figure 1 (middle), HOLEX visually first forms the full tensor product of $\boldsymbol{a}$ and $\boldsymbol{b}$, then multiplies each row with the corresponding dimension in $\boldsymbol{c}$, and finally sums up along each (circular) diagonal line. Computationally, HOLEX continues to benefit from the use of fast Fourier transform: $h(\boldsymbol{a}, \boldsymbol{b}; \boldsymbol{c}) = \mathcal{F}^{-1}\left(\overline{\mathcal{F}(\boldsymbol{c} \circ \boldsymbol{a})} \circ \mathcal{F}(\boldsymbol{b})\right)$.

On one hand, HOLEX falls back to HOLE if we only use one perturbation vector with all non-zero entries. This is because one can always rescale $\boldsymbol{a}$ to subsume the effect of $\boldsymbol{c}$.

On the other hand, we can expand HOLE to more complex models by using multiple perturbation vectors. Suppose we have $l$ vectors $\boldsymbol{c}_0, \ldots, \boldsymbol{c}_{l-1}$. The rank-$l$ HOLEX is defined as the concatenation of the perturbed holographic embeddings induced by $\boldsymbol{c}_0, \ldots, \boldsymbol{c}_{l-1}$, i.e.,

$$h(\boldsymbol{a}, \boldsymbol{b}; \boldsymbol{c}_0, \ldots, \boldsymbol{c}_{l-1}) = [h(\boldsymbol{a}, \boldsymbol{b}; \boldsymbol{c}_0), h(\boldsymbol{a}, \boldsymbol{b}; \boldsymbol{c}_1), \ldots, h(\boldsymbol{a}, \boldsymbol{b}; \boldsymbol{c}_{l-1})]. \tag{8}$$

For simplicity of notation, let matrix $\boldsymbol{C}_l$ denote $(\boldsymbol{c}_0, \ldots, \boldsymbol{c}_{l-1})$ and write $h(\boldsymbol{a}, \boldsymbol{b}; \boldsymbol{C}_l)$ to represent $h(\boldsymbol{a}, \boldsymbol{b}; \boldsymbol{c}_0, \ldots, \boldsymbol{c}_{l-1})$. Treating each $h(\boldsymbol{a}, \boldsymbol{b}; \boldsymbol{c}_i)$ as a column vector, the entire expanded embedding, $h(\boldsymbol{a}, \boldsymbol{b}; \boldsymbol{C}_l)$, is a $d \times l$ matrix. For the tail-prediction task (analogously for head-prediction), the final rank-$l$ HOLEX score for a triple $(s, r, o)$ is defined as:

$$\sigma(\boldsymbol{s}, \boldsymbol{r}, \boldsymbol{o}) = \sum_{j=0}^{l} h(\boldsymbol{s}, \boldsymbol{r}; \boldsymbol{c}_j) \cdot \boldsymbol{o}. \tag{9}$$

Importantly, this expanded embedding has the *same number of parameters* as HOLE itself.[3]

We start with a basic question: *Does rank-$l$ HOLEX capture more information than rank-$l'$ when $l > l'$?* The answer is affirmative if $c_0, \ldots, c_{l-1}$ are linearly independent. In fact, Theorem 1 shows that under this setting, rank-$d$ HOLEX is equivalent to the full tensor product up to a linear transformation.

**Theorem 1.** *Let $a, b \in \mathbb{R}^d, l = d$, and $R$ be the matrix of the full tensor product matrix arranged according to diagonal lines, i.e., $R_{i,j} = a_i b_{(i+j) \bmod d}$. Then rank-$d$ HOLEX satisfies:*

$$h(a, b; C_d) = R^{\mathrm{T}} C_d.$$

Note that this linear transformation is invertible if $C_d$ has full rank. In other words, learning a rank-$d$ expanded holographic embedding is equivalent to learning the full tensor product. As an example, consider the RESCAL model with score function $r^{\mathrm{T}}(s \otimes o)$. This is an inner product between the relation embedding $r$ and the full tensor product matrix $(s \otimes o)$ between the subject and object entities. Suppose we replace the tensor product matrix $(s \otimes o)$ with the full expanded holographic embedding $h(s, o; C_d)$, obtaining a new model $r^{\mathrm{T}} h(s, o; C_d)$. Theorem 1 states that the original tensor product matrix $(s \otimes o)$ is connected to $h(s, o; C_d)$ via a linear transformation, making the two embedding models, $r^{\mathrm{T}}(s \otimes o)$ and $r^{\mathrm{T}} h(s, o; C_d)$, essentially equivalent.

## 3.1 Low Rank Holographic Expansions

Theorem 1 states that if we could afford $d$ perturbation vectors, then HOLEX is equivalent to the full tensor product (RESCAL) matrix. *What happens if we cannot afford all $d$ perturbation vectors?* We will see that, in this case, HOLEX forms a collection of models with increasingly richer representation power and correspondingly higher computational needs. Our goal is to choose a family of perturbation vectors that provides a substantial benefit even if only a handful of vectors are used in HOLEX.

Different choices of linearly independent families of perturbation vectors extract different information from the full RESCAL matrix, thereby leading to different empirical performance. For example, consider using the truncated identity matrix $I_{k \times d}$, i.e., the first $k$ columns of the $d \times d$ identity matrix, as the perturbation vectors. This is equivalent to retaining the first $k$ major diagonal lines of the full RESCAL matrix and ignoring everything else in it. Empirically, we found that using $I_{k \times d}$ substantially *worsened* performance. Our intuition is that such a choice is worse than using perturbation vectors that condense information from the *entire* RESCAL matrix, i.e., vectors with a wider footprint. We consider two such perturbation families, Haar and random 0/1 vectors.

The following example illustrates the intuition behind such wide-footprint vectors being a better fit for the task than $I_{k \times d}$. Consider the tuple $\langle$Alice, review, paper42$\rangle$. When we embed Alice and paper42 as entities, it may result in the $i$-th embedding dimension being indicative of a person (i.e., this dimension has a large value whenever the entity is a person) and the $j$-th dimension being indicative of an article. In this case, the $(i, j)$-th entry of the interaction matrix will have a large value for the pair $\langle$Alice, paper42$\rangle$, signaling that it fits the relation "review". If the $(i, j)$-th entry is far away from the main diagonal, it will be zeroed out (thus losing the information) when using $I_{k \times d}$ for perturbation, but captured by vectors with a wide footprint.

### 3.1.1 Perturbation with Low Frequency Haar Vectors

The Haar wavelet system [6, 10] is widely used in signal processing. The $2 \times 2$ Haar matrix $H_2$ associated with the Haar wavelet is shown on the left in Figure 2, which also shows $H_4$. In general, the $2n \times 2n$ Haar matrix $H_{2n}$ can be derived from the $n \times n$ Haar matrix $H_n$ as shown on the right in Figure 2, where $\otimes_k$ represents the Kronecker product and $I$ the identity matrix.

Haar matrices have many desirable properties. Consider multiplying $H_4$ with a vector $a$. We can see that the inner product between the first row of $H_4$ and $a$ gives the sum of the entries of $a$ (i.e., $\sum_{i=0}^{3} a_i$). The inner product between the second row of $H_4$ and $a$ gives the difference between the

$$\boldsymbol{H}_2 = \begin{pmatrix} 1 & 1 \\ 1 & -1 \end{pmatrix} \qquad \boldsymbol{H}_4 = \begin{pmatrix} 1 & 1 & 1 & 1 \\ 1 & 1 & -1 & -1 \\ 1 & -1 & 0 & 0 \\ 0 & 0 & 1 & -1 \end{pmatrix} \qquad \boldsymbol{H}_{2n} = \begin{pmatrix} \boldsymbol{H}_n & \otimes_k & [1,1] \\ \boldsymbol{I}_n & \otimes_k & [1,-1] \end{pmatrix}$$

Figure 2: Haar matrices of order 2, 4, and $2n$.

sum of the first half of $\boldsymbol{a}$ and the second half ($\sum_{i=0}^{1} a_i - \sum_{i=2}^{3} a_i$). Importantly, *we can infer the sum of each half of $\boldsymbol{a}$ by examining these two inner products.* Generalizing this, consider the first $2^k$ rows of $\boldsymbol{H}_d$, referred to as the (unnormalized) $2^k$ Haar wavelets with the lowest frequency. If we split vector $\boldsymbol{a}$ into $2^k$ segments of equal size, then *one can infer the partial sum of each segment by computing the inner product of $\boldsymbol{a}$ with the first $2^k$ rows of $\boldsymbol{H}_d$.*

This view provides an intuitive interpretation of HOLEX when using perturbation with low frequency Haar vectors. In fact, we can prove that HOLEX using the first $2^k$ rows of $\boldsymbol{H}_d$ yields an embedding that contains the partial sums of $2^k$ equal-sized segments along each (circular) diagonal line of the tensor product matrix. This is stated formally in Proposition 1. The case of using the first two rows of $\boldsymbol{H}_d$ is visually depicted in the rightmost panel of Figure 1.

**Proposition 1.** *Let $1 \le k \le K, d = 2^K, l = 2^k$, $\boldsymbol{H}_l$ and $\boldsymbol{H}_d$ be Haar matrices of size $l$ and $d$, respectively, and $\boldsymbol{H}_{d,k}$ be a matrix that contains the first $l$ rows of $\boldsymbol{H}_d$. $h(\boldsymbol{a}, \boldsymbol{b}; \boldsymbol{H}'_{d,k})$ is the compositional operator for HOLEX using $\boldsymbol{H}_{d,k}$ as perturbation vectors. Let $\boldsymbol{R}$ be the full tensor product matrix arranged according to diagonal lines, i.e., $\boldsymbol{R}_{i,j} = a_i b_{(i+j) \bmod d}$. Define:*

$$\boldsymbol{W} = \frac{1}{l} \boldsymbol{H}_l^{\mathrm{T}} \, h(\boldsymbol{a}, \boldsymbol{b}; \boldsymbol{H}_{d,k}^{\mathrm{T}})$$

*Then, $\boldsymbol{W}$ captures the partial column sums of $\boldsymbol{R}$. In other words, $\boldsymbol{W}_{i,j}$ is the sum of entries from $\boldsymbol{R}_{di/l,j}$ to $\boldsymbol{R}_{d(i+1)/l-1,j}$, where all indices start from 0.*

Proposition 1 formalizes how HOLEX forms an interpolation between HOLE and the full tensor product as an increasing number of Haar wavelets is used as perturbation vectors. While HOLE captures the sum along each full diagonal line, HOLEX gradually enriches the representation by adding subsequence sums as we include more and more rows from the Haar matrix.

### 3.2 Projection with Random 0/1 Vectors

We next consider random perturbation vectors, each of whose entries is sampled independently and uniformly from $\{0, 1\}$. As suggested by Figure 1 (middle), HOLE perturbed with one such random 0/1 vector is equivalent to randomly zeroing out roughly half the $d$ rows (corresponding to the 0s in the vector) from the tensor product matrix, before summing along each (circular) diagonal line.

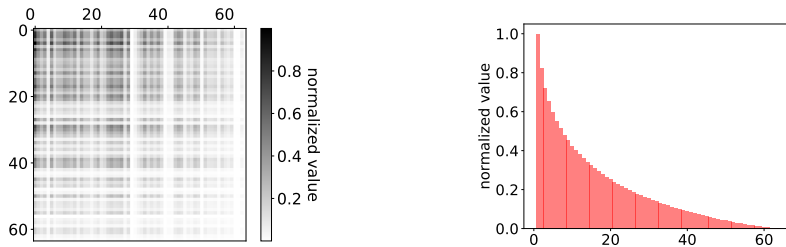

Figure 3: Sparse nature of full $64 \times 64$ RESCAL matrices learned from the FB15k dataset. The heat map on the left shows a typical entity-relation matrix, i.e., a particular $\boldsymbol{s} \otimes \boldsymbol{r}$. The plot on the right shows the average magnitudes of the entries in each (circular) diagonal line, normalized so that the largest entry in each diagonal is 1, sorted in decreasing order, and averaged over the entire dataset.

Random vectors work particularly well if the full tensor product matrix is *sparse*, which turns out to often be the case. Figure 3 illustrates this sparsity for the FB15K dataset. The heat map on the left highlights that there are relatively few large (dark) entries overall. The plot on the right shows that each circular diagonal line, on average, is dominated by very few large entries. For example, on

average, the 5$^{\text{th}}$ largest entry (out of 64) has a magnitude of only about half that of the largest. The values decay rapidly. This is in line with our expectation that different entries of the RESCAL matrix carry different semantic information, not all of which is generally relevant for all entity-relation pairs.

To understand why random vectors are suitable for sparse interactions, consider the extreme but intuitive case where only *one* of the $d$ entries in each diagonal line has large magnitude, and the rest are close to zero. For a particular diagonal line, one random vector zeros out a set of approximately $d/2$ entries, and the second random vector zeros out another set of $d/2$ entries, chosen randomly and independently of the first set. The number of entries that are not zeroed out by either of the two random vectors is thus approximately $d/4$. Continuing this reasoning, in expectation, only one entry will "survive", i.e., remain not zeroed out, if one adds $\log_2 d$ of such 0/1 random vectors.

Suppose we apply HOLEX with $2 \log d$ random vectors. For a particular diagonal line, approximately half ($\log d$) of the random vectors will zero out the unique row of the large entry, thereby resulting in a small sum for that diagonal. Consider those $\log d$ random vectors that produce large sums. According to the previous reasoning, there is, in expectation, only one row that none of these vectors zeros out. The intersection of this row and the diagonal line must, then, be the location of the large entry.

Therefore, we have the following theorem, saying that if there is only one non-zero entry in every diagonal, HOLEX can recover the whole matrix.

**Theorem 2.** *Suppose there is only one non-zero entry, of value 1, in each diagonal line of the tensor product matrix. Let $\eta > 0$ and $d$ be the embedding dimension.* HOLEX *expanded with $\lceil 3 \log d - \log \eta \rceil - 1$ random 0/1 vectors can locate the non-zero entry in each diagonal line of the tensor product matrix with probability at least $1 - \eta$.*

Assuming exactly one non-zero entry per diagonal might be too strong, but it can be weakened using techniques from compressed sensing, as reflected in the following theorem:

**Theorem 3.** *Suppose each diagonal line of the tensor product matrix is $s$-sparse, i.e., has no more than $s$ non-zero entries. Let $\boldsymbol{A} \in \mathbb{R}^{l \times d}$ be a random 0/1 matrix. Let $\eta \in (0, 1)$ and $l \geq C(s \log(d/s) + \log(\epsilon^{-1}))$ for a universal constant $C > 0$. Then* HOLEX *with the rows of $\boldsymbol{A}$ as perturbation vectors can recover the tensor product matrix, i.e., identify all non-zero entries, with probability at least $1 - \eta$.*

The proofs of the above two theorems are deferred to the Appendix. We note that Theorem 3 also holds in the noisy setting where diagonal lines have $s$ large entries, but are corrupted by some bounded noise vector $e$. In this case, we do not expect to fully recover the original tensor product matrix, but can identify a matrix that is close enough, which is sufficient for machine learning applications. We omit the details (cf. Theorem 2.7 of Rauhut [20]). Thus, HOLEX works provably as long as each diagonal of the tensor product matrix can be *approximated* by a sparse vector.

## 4   Experiments

For evaluation, we use the standard knowledge completion dataset FB15K [5]. This dataset is a subset of Freebase [3], which contains a large number of general facts about the world. FB15K contains 14,951 entities, 1,345 relations, and 592,213 facts. The facts are divided into 483,142 for training, 50,000 for validation, and 59,071 for testing.

We follow the evaluation methodology of prior work in this area. For each triple $(s, r, o)$, we create a head prediction query $(?, r, o)$ and a tail prediction query $(s, r, ?)$. For head prediction (tail prediction is handled similarly), we use the knowledge completion method at hand to rank all entities based on their predicted likelihood of being the correct head, resulting in an ordered list $L$. Since many relations are not 1-to-1, there often are other (already known) valid facts of the form $(s', r, o)$ with $s' \neq s$ in the training data. To account for these equally valid answers, we follow prior work and filter out such other valid heads from $L$ to obtain $L'$. Finally, for this query, we compute three metrics: the 0-based rank $r$ of $s$ in this (filtered) ordered list $L'$, the reciprocal rank $\frac{1}{r+1}$, and whether $s$ appears among the top $k$ items in $L'$ (HITS@$k$, for $k \in \{10, 5, 1\}$). The overall performance of the method is taken to be the average of these metrics across all head and tail prediction queries.

We reimplemented HOLE using the recent framework of Shi and Weninger [23], which is based on TensorFlow [1] and is optimized for multiple CPUs. We consider both the original embedding dimension of 150, and a larger dimension of 256 that is better suited for our Haar vector based linear

perturbation. In the notation of Shi and Weninger [23], we changed their interaction component between entity $e$ and relation $r$ from $e \oplus r = D_e e + D_r r + b_c$ to the (expanded) holographic interaction $h(e, r)$. We also dropped their non-linearity function, $\tanh$, around this interaction for slightly better results. Their other implementation choices were left intact, such as computing interaction between $e$ and $r$ rather than between two entities, using dropout, and other hyper-parameters.

## 4.1 Impact of Varying the Number of Perturbation Vectors

To gain insight into HOLEX, we first consider the impact of adding an increasing number $l$ of linear perturbation vectors. We start with a small embedding dimension, 32, which allows for a full interpolation between HOLE and RESCAL. We then report results with embedding dimension 256, with up to 8 random 0/1 vectors. Figure 4 depicts the resulting HITS@10 and Mean Rank metrics, as well as the training time per epoch (on a 32-CPU machine on Google Cloud Platform).

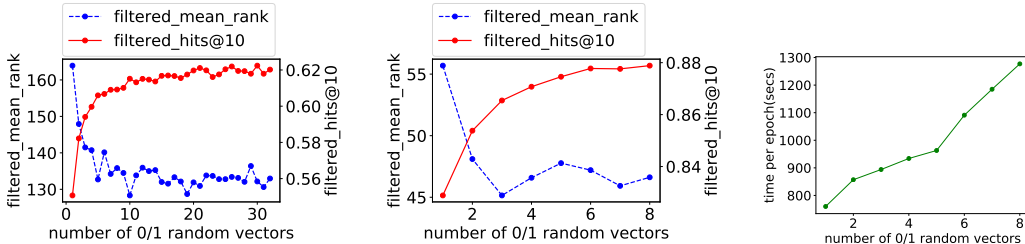

Figure 4: Impact of using a varying number $l$ of random 0/1 vectors on the performance of HOLEX on FB15K. Left: HITS@10 and Mean Rank for a full interpolation between HOLE ($l = 1$) and RESCAL ($l = 32$), when the embedding dimension $d = 32$. Middle: Similar trend for $d = 256$. Right: Training time per epoch for experiments with $d = 256$.

On both small- and large-scale experiments, we observe that both the mean rank and HIT@10 metrics generally improve (ignoring random fluctuations) as $l$ increases. On the large-scale experiment, even with $l = 2$, we already observe a substantial, 2.5% improvement in HITS@10 (higher is better) and a reduction in mean rank (lower is better) by 8. In this particular case, mean rank saturates after $l = 3$, although HITS@10 continues to climb steadily till $l = 8$, and suggests further gains if even more perturbation vectors were to be used. The rightmost plot indicates that the training time scales roughly linearly, thus making $l$ an effective knob for trading off test performance with training time.

## 4.2 Comparison with Existing Methods

We now compare HOLEX with several representative baselines. All baseline numbers, except for our reimplementation of HOLE, are taken from Shi and Weninger [23], who also report performances of additional baselines, which fared worse than TransR reported here.

**Remark 1.** *In private communication, Shi and Weninger noted that the published numbers for their method, ProjE, were inaccurate due to a bug (Github issue #3). We use their updated code from https://github.com/bxshi/ProjE and new suggested parameters, reported here for completeness: max 50 iterations, learning rate 0.0005, and negative sampling weight 0.1. We increased the embedding dimension from 200 to 256 for consistency with our method and reduced batch size to 128, which improved the HITS@10 metric for the best variant, ProjE_listwise, from 80.0% to 82.9%. We use this final number here as the best ProjE baseline.*

Table 1 summarizes our main results, with various method sorted by increasing HITS@10 performance. The best baselines numbers are highlighted in bold, and so are the best numbers using our expanded holographic embeddings method. We make a few observations.

First, although the RESCAL approach [18], which works with the full outer product matrix, is capable of capturing rich correlation by looking at every pair of dimensions, the resulting quadratically many parameters make it difficult to train in practice, eventually resulting in poor performance.

Second, models such as TransE [5] and TransR [16] that rely on simple vector arithmetic such as adding/subtracting vectors, are unable to capture rich correlation, again resulting in low performance.

| Knowledge Completion Method | Mean Rank | HITS@10 (%) | MRR | HITS@5 (%) | HITS@1 (%) |
|---|---|---|---|---|---|
| EXISTING METHODS | | | | | |
| RESCAL [18] | 683 | 44.1 | - | - | - |
| TransE [5] | 125 | 47.1 | - | - | - |
| TransR [16] | 77 | 68.7 | - | - | - |
| TransE + Rev [15] | 63 | 70.2 | - | - | - |
| HOLE (original, dim=150) [19] | - | 73.9 | 0.524 | - | 40.2 |
| HOLE (reimplementation, dim=150) | 70 | 78.4 | 0.588 | 72.0 | 47.7 |
| ProjE_pointwise* (dim=256) [23] | 71 | 80.2 | 0.650 | 74.8 | 56.7 |
| ProjE_wlistwise* (dim=256) [23] | 64 | 82.1 | 0.666 | 76.8 | 57.9 |
| ProjE_listwise* (dim=256) [23] | 53 | 82.9 | 0.665 | 78.1 | 56.8 |
| HolE (reimplementation, dim=256) | 51 | 83.0 | 0.665 | 77.9 | 56.9 |
| ComplEx [27] | - | 84.0 | 0.692 | - | 59.9 |
| PTransE (ADD, len-2 path) [15] | 54 | 83.4 | - | - | - |
| PTransE (ADD, len-3 path) [15] | 58 | 84.6 | - | - | - |
| DistMult [29], re-tuned by Kadlec et al. [13] | **42** | **89.3** | **0.798** | - | - |
| PROPOSED METHOD (dim=256) | | | | | |
| HolE (reimplemented baseline from above) | 51 | 83.0 | 0.665 | 77.9 | 56.9 |
| HOLEX, 8 Haar vectors | 51 | 86.7 | - | - | - |
| HOLEX, 2 random 0/1 vectors | 48 | 85.4 | 0.720 | 81.4 | 64.0 |
| HOLEX, 4 random 0/1 vectors | 47 | 87.1 | 0.763 | 83.9 | 69.8 |
| HOLEX, 8 random 0/1 vectors | **47** | 87.9 | 0.786 | 85.0 | 73.1 |
| HOLEX, 16 random 0/1 vectors | 49 | **88.6** | **0.800** | **86.0** | **75.0** |

Table 1: Expanded holographic embeddings, HOLEX, outperform a variety of knowledge completion methods on the FB15K dataset. Mean Rank (0-based) and HITS@10 are the main metrics we track; other metrics are reported for a more comprehensive comparison with prior work. Numbers are averages across head- and tail-prediction tasks, individual results for which may be found in the Appendix. * See Remark 1 for an explanation of ProjE results, and Footnote 4 for very recent models.

Third, reimplementing HOLE using the ProjE framework increases HITS@10 from 73.9% to 78.4%, likely due to improved training with the TensorFlow backend, regularization techniques like dropout, and entity-relation interaction rather than original HOLE's entity-entity interaction. Further, simply increasing the embedding dimension from 150 to 256 allows HOLE to achieve 83.0% HITS@10, higher than most baseline methods that do not explicitly model KG paths, except for DistMult [29] which was re-tuned very carefully for this task [13] to achieve state-of-the-art results.[4]

Relative to the (reimplemented) HOLE baseline, our proposed HOLEX with 8 Haar vectors improves the HITS@10 metric by 3.7%. The use of random 0/1 vectors appears somewhat more effective, achieving 88.6% HITS@10 with 16 such vectors, which is a 5.7% improvement over ProjE, which formed our codebase. This setting also achieves a mean reciprocal rank (MRR) of 0.800 and HITS@1 of 75.0%, matching or outperforming a wide variety of existing methods along various metrics.[5]

## 5 Conclusion

We proposed expanded holographic embeddings (HOLEX), a new family of embeddings for knowledge graphs that smoothly interpolates between the full product matrix of correlations on one hand, and an effective lower dimensionality method, namely HOLE, on the other. By concatenating several linearly perturbed copies of HOLE, our approach allows the system to focus on different subspaces of the full embedding space, resulting in a richer representation. It recovers the full interaction matrix when sufficiently many copies are used. Empirical results on the standard FB15K dataset demonstrate the strength of HOLEX even with only a handful of perturbation vectors, and the benefit of being able to select a point that effectively trades off expressivity of relational embeddings with computation.

## Footnotes

[1] A composition operator can be between two entities, or between an entity and a relation.

[2] As discussed later, our improved reimplementation of HOLE, inspired by recent work [23], uses a slight variation for the tail-prediction (a.k.a. object-prediction) task, namely $\sigma(\boldsymbol{s}, \boldsymbol{o}, \boldsymbol{r}) = [\boldsymbol{s} \star \boldsymbol{r}] \cdot \boldsymbol{o}$. Analogously for head-prediction (a.k.a. subject-prediction).

[3]While we discuss expansion in the context of HOLE, it is evident from Eq. (9) that one can easily generalize the notion (even if not the theoretical results that follow) to *any* embedding method that can be decomposed as $\sigma(s, r, o) = g(f(s, r), o)$, or a similar decomposition for another permutation of $s, r, o$. In this case, the expanded version would simply be $\sum_{j=0}^{l} g(f(c_j \circ s, r), o)$.

[4] A recent model called EKGN [22] outperforms this with HITS@10 at 92.7% and mean rank 38. Explicitly modeling reciprocal relations, adding a new regularizer, and using weighted nuclear 3-norm have also recently been shown to improve both CP decomposition and ComplEx to HITS@10 of 91% and MRR 0.86, albeit with much larger embedding dimensions of 4,000 and 2,000 for CP and ComplEx, respectively [14, 21].

[5] Our HOLEX implementation uses the hyper-parameters recommended for ProjE, except for embedding dimension 256 and batch size 128. Hyper-parameter tuning targeted for HOLEX should improve results further.

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
