[Supplementary Material]

# A    Additional Empirical Results

Tables 2 and 3 summarize the performance of HOLEX for the head- and tail-prediction tasks, respectively. Note that the corresponding numbers are averaged when reporting the main results in Table 1 on the full task.

As has been observed in prior work, the tail-prediction task is considerably easier than head-prediction for named-entity knowledge bases such as Freebase. This is because many-to-one relations tend to be more common than one-to-many relations. For instance, many people "live in" one city or "work for" one company; where as relatively few people have been the "president of" the United States).

We see, for example, that when using 8 random 0/1 vectors in HOLEX, the tail-prediction HITS@10 metric is 90.5%, which is 5.2% higher than that for head-prediction. Similarly, the mean rank for tail-prediction is 35 in this case, compared to 58 for head prediction.

| Knowledge Completion Method | Mean Rank | HITS@10 (%) | MRR | HITS@5 (%) | HITS@1 (%) |
|---|---|---|---|---|---|
| HolE (reimplemented baseline, dim=256) | 62 | 80.3 | 0.640 | 75.1 | 54.6 |
| HOLEX, 8 Haar vectors | 63 | 84.1 | - | - | - |
| HOLEX, 2 random 0/1 vectors | 60 | 82.8 | 0.696 | 78.7 | 61.8 |
| HOLEX, 4 random 0/1 vectors | 59 | 84.6 | 0.740 | 81.4 | 67.7 |
| HOLEX, 8 random 0/1 vectors | 58 | 85.3 | 0.763 | 82.5 | 70.9 |
| HOLEX, 16 random 0/1 vectors | 61 | 86.1 | 0.777 | 83.4 | 72.8 |

Table 2: Performance of HOLEX on the head-prediction task. Table 1 reports the average of this and tail-prediction performance.

| Knowledge Completion Method | Mean Rank | HITS@10 (%) | MRR | HITS@5 (%) | HITS@1 (%) |
|---|---|---|---|---|---|
| HolE (reimplemented baseline, dim=256) | 41 | 85.6 | 0.690 | 80.7 | 59.2 |
| HOLEX, 8 Haar vectors | 39 | 89.3 | - | - | - |
| HOLEX, 2 random 0/1 vectors | 36 | 88.0 | 0.744 | 84.1 | 66.3 |
| HOLEX, 4 random 0/1 vectors | 35 | 89.5 | 0.785 | 86.5 | 72.0 |
| HOLEX, 8 random 0/1 vectors | 35 | 90.5 | 0.810 | 87.5 | 75.4 |
| HOLEX, 16 random 0/1 vectors | 37 | 91.1 | 0.823 | 88.6 | 77.2 |

Table 3: Performance of HOLEX on the tail-prediction task. Table 1 reports the average of this and head-prediction performance.

# B    Proof Details

*Proof of Theorem 1.* According to the definition of the expanded holographic embedding. We have the $j, i$-th entry of the matrix $h(\boldsymbol{a}, \boldsymbol{b}; \boldsymbol{C}_d)$ is:

$$[h(\boldsymbol{a}, \boldsymbol{b}; \boldsymbol{C}_d)]_{j,i} = \sum_{l=0}^{d-1} c_{i,l} a_l b_{(l+j) \bmod d}.$$

in which $c_{i,l}$ is the $l, i$-th entry of the matrix $\boldsymbol{C}_d$, and $a_l b_{(l+j) \bmod d}$ is $\boldsymbol{R}_{l,j}$ – the $l, j$-th entry of matrix $\boldsymbol{R}$. Therefore,

$$h(\boldsymbol{a}, \boldsymbol{b}; \boldsymbol{C}_d)' = \boldsymbol{C}_d' \boldsymbol{R}.$$

which is equivalent to what the Theorem states.    $\square$

**Definition 1.** *A random 0/1 matrix $\boldsymbol{A} \in \{0,1\}^{l \times d}$ is a matrix whose entries are chosen independently and uniformly at random from $\{0, 1\}$.*

**Claim 1.** *Suppose $\boldsymbol{x}, \boldsymbol{y} \in \mathbb{R}^d$ are two vectors, each with exactly one non-zero entry, and at different locations. Let $\boldsymbol{A} \in \{0,1\}^{l \times d}$ be a random 0/1 matrix. Then $\Pr(\boldsymbol{A}\boldsymbol{x} = \boldsymbol{A}\boldsymbol{y}) \leq \frac{1}{2^l}$.*

*Proof.* Suppose the $i$-th entry is the unique non-zero in $\boldsymbol{x}$, and similarly for the $j$-th entry in $\boldsymbol{y}$. $\boldsymbol{A}\boldsymbol{x} = \boldsymbol{A}\boldsymbol{y}$ must imply that $\boldsymbol{A}(:, i) = \boldsymbol{A}(:, j)$. Otherwise, suppose $\boldsymbol{A}_{k,i} = 1$ but $\boldsymbol{A}_{k,j} = 0$, this leads to $\boldsymbol{A}\boldsymbol{x}$ to be non-zero but $\boldsymbol{A}\boldsymbol{y}$ to be zero. Contradiction. Given this fact,

$$\Pr(\boldsymbol{A}\boldsymbol{x} = \boldsymbol{A}\boldsymbol{y}) \leq \Pr(\boldsymbol{A}(:, i) = \boldsymbol{A}(:, j)) = 1/2^l$$

as claimed. □

*Proof of Theorem 2.* Because $d$ diagonal lines are mutually independent, it suffices to prove the statement holds for one diagonal line with probability at least $1 - \eta/d$. A union bound argument can be applied to show that the statement holds for all $d$ diagonal lines with probability at least $1 - \eta$. In this case, the rest of the proof focuses on one diagonal line.

The effect of applying expanded holographic embedding with $l$ random 0/1 vectors on one diagonal line is to multiply this diagonal line with a $l$-by-$d$ random 0/1 matrix $\boldsymbol{A}$. This fact can be quickly checked with the graphical example in Figure 1 (middle). Suppose $\boldsymbol{x}$ and $\boldsymbol{y}$ are two possible configurations of one diagonal line of interest (i.e., both $\boldsymbol{x}$ and $\boldsymbol{y}$ have one non-zero entry of value 1). If a random 0/1 matrix $\boldsymbol{A}$ can tell apart every pairs of $\boldsymbol{x}$ and $\boldsymbol{y}$, we can decide which configuration the diagonal line is actually in by examining the result of the expanded holographic embedding. In other words, it is sufficient to prove the following: *let $l = \lceil 3 \log d - \log \eta \rceil - 1$. sample an $l$-by-$d$ random 0/1 matrix $\boldsymbol{A}$, then with probability at least $1 - \eta/d$, we must have $\boldsymbol{A}\boldsymbol{x} \neq \boldsymbol{A}\boldsymbol{y}$ holds, for any two vectors $\boldsymbol{x}$ and $\boldsymbol{y}$ with exact one non-zero entry of value 1.*

$$\Pr(\forall \boldsymbol{x}, \boldsymbol{y} \in D : \boldsymbol{x} \neq \boldsymbol{y}, \boldsymbol{A}\boldsymbol{x} \neq \boldsymbol{A}\boldsymbol{y}) \tag{10}$$

$$= 1 - \Pr(\exists \boldsymbol{x}, \boldsymbol{y} \in D : \boldsymbol{x} \neq \boldsymbol{y}, \boldsymbol{A}\boldsymbol{x} = \boldsymbol{A}\boldsymbol{y}) \tag{11}$$

$$\geq 1 - \frac{d(d-1)}{2} Pr(\boldsymbol{A}\boldsymbol{x}_0 = \boldsymbol{A}\boldsymbol{y}_0) \tag{12}$$

$$\geq 1 - \frac{d(d-1)}{2} \frac{1}{2^l} \geq 1 - \eta/d. \tag{13}$$

Here, $D$ is the space with vectors of exact one non-zero entry of value 1. The size of $D$ is $\frac{d(d-1)}{2}$. It is a union bound argument from (2) to (3). From (3) to (4) we use Claim 1. The last inequality is because $l \geq 3 \log d - \log \eta - 1$. □

The proof of theorem 3 makes many connections to compressed sensing. We provide a brief review here. Many definitions and lemmas can be found in [20]. We first introduce the notion of restricted isometry property.

**Definition 2** (restricted isometry property [20]). *The restricted isometry constant $\delta_s$ of a matrix $\boldsymbol{A} \in \mathbb{R}^{m \times d}$ is defined as the smallest $\delta_s$ such that*

$$(1 - \delta_s)\|x\|_2^2 \leq \|\boldsymbol{A}x\|_2^2 \leq (1 + \delta_s)\|x\|_2^2$$

*for all $s$-sparse $x \in \mathbb{R}^d$.*

It is well known that restricted isometry property implies recovery of sparse vectors, which can be shown below.

**Lemma 1** (Theorem 2.6, [20]). *Suppose the restricted isometry constants $\delta_{2s}$ of a matrix $\boldsymbol{A} \in \mathbb{R}^{m \times d}$ satisfies $\delta_{2s} < \frac{1}{3}$, then every $s$-sparse vector $x^* \in \mathbb{R}^d$ is recovered by $\ell_1$-minimization.*

Therefore, in order to guarantee sparse recovery of $x^*$, we need a good matrix $\boldsymbol{A}$. It turns out that random Bernoulli matrix has good restricted isometry constant upper bound:

**Lemma 2** (Theorem 2.12, [20]). *Let $\boldsymbol{A} \in \mathbb{R}^{m \times d}$ be a Bernoulli random matrix, where every entry of the matrix takes the value $\frac{1}{\sqrt{m}}$ or $-\frac{1}{\sqrt{m}}$ with equal probability. Let $\epsilon, \delta \in (0, 1)$ and assume $m \geq C\delta^{-2}(s \log(d/s)) + \log(\epsilon^{-1})$ for a universal constant $C > 0$. Then with probability at least $1 - \epsilon$ the restricted isometry constant of $\boldsymbol{A}$ satisfies $\delta_s \leq \delta$.*

**Lemma 3** (Compressed sensing). *Let $\boldsymbol{A} \in \mathbb{R}^{m \times d}$ be a Bernoulli random matrix, where every entry of the matrix takes the value $\frac{1}{\sqrt{m}}$ or $-\frac{1}{\sqrt{m}}$ with equal probability. Let $x^* \in \mathbb{R}^d$ be a vector with at most $s$ non-zero entries. let $\epsilon \in (0,1)$ and assume*

$$m \geq C(s \log(d/s) + \log(\epsilon^{-1}))$$

*for a universal constant $C > 0$. Let random linear measurements $y = \boldsymbol{A}x^*$ be given, and $x$ be a solution of*

$$\min_z \|z\|_1 \qquad \text{subject to} \quad y = \boldsymbol{A}z \qquad (14)$$

*Then with probability at least $1 - \epsilon$, $x = x^*$.*

*Proof of Lemma 3.* By setting $\delta = \frac{1}{3}$ in Lemma 2, and using Lemma 1, Lemma 3 is proved. $\qquad\square$

**Lemma 4.** *Let $\epsilon \in (0,1)$. If $x_1, x_2 \in \mathbb{R}^d$ have at most $s$ non-zero entries, $\boldsymbol{A} \in \mathbb{R}^{m \times d}$ is a Bernoulli random matrix, $m \geq C(s \log(d/s) + \log(\epsilon^{-1}))$ for a universal constant $C > 0$. If we have $y_1 = \boldsymbol{A}x_1$, $y_2 = \boldsymbol{A}x_2$, and $y_1 = y_2$, then with probability at least $1 - \epsilon$, we know that $x_1 = x_2$.*

*Proof.* Lemma 4 is a corollary of Lemma 3. Lemma 3 says that if $x$ is sparse, then $y$ uniquely determines $x$ by running $\ell_1$ regression. That means, $y$ can be used as a certificate for testing whether the unknown vector $x$ is what we want. Using Lemma 3, we know that by running $\ell_1$ regression, we could recover the unique solution for both $y_1 = \boldsymbol{A}x_1$ and $y_2 = \boldsymbol{A}x_2$. Since $y_1 = y_2$, by probability $1 - \epsilon$, the two programs have the same unique solution, denoted as $x'$.

If $x_1 \neq x_2$, it means $x'$ is not the same as at least one of them. Without loss of generality, assume $x' \neq x_1$. This contradicts the claim of Theorem 3, which says $x'$ equals $x_1$. $\qquad\square$

*Proof of Theorem 3.* Theorem 3 is a simple corollary of Lemma 4. To prove Theorem 3, it is sufficient to prove that a $l$-by-$d$ Bernoulli random matrix can differentiate all $s$-sparse vectors with high probability, which is implied by Lemma 4. $\qquad\square$