[Reviews · NeurIPS 2018]

Reviewer 1



** Summary The authors present a new scoring function for neural link prediction that allows one to interpolate between expressivity/complexity and computational tractability. They define a model which generalises an existing low-complexity model HolE by stacking a number of instances of HolE, each perturbed with a perturbation vector c. The authors show how, for an appropriately chosen set of c vectors, this model is equivalent to RESCAL, a high-complexity model. They provide a number of theoretical results characterising their model for two different classes of perturbation vectors. Finally, they demonstrate that their model improves on existing methods on the FB15K dataset. ** Quality The work presented here seems to be theoretically and empirically sound. I do have one comment regarding their choice of benchmark dataset. A recent paper [1] showed how one can get quite good performance on FB15K using a trivial model, suggesting it might not be such a good benchmark for link prediction. The alternative dataset FB15K-237 does not have this issue. I would recommend the authors re-run their experiments with FB15K and either replace the results in the paper, or add a new results table. Knowledge graph completion papers also tend to evaluate on WN18 and WN18RR, e.g. see Table 2 in [2] ** Clarity The paper is very clearly written and was a pleasure to read On line 115, the authors state RESCAL becomes infeasible for modestly sized KGs. RESCAL scales poorly with the number of relations, so it would be more accurate to say RESCAL becomes infeasible for KGs with a modestly large number of relations. RESCAL will scale no worse than other methods on graphs with a large number of entities but a relatively small number of relations ** Novelty The model presented is a non-trivial generalisation of existing methods and the theoretical results provide useful insight. ** Significance I think this is a good contribution to the link prediction literature, providing a nice way to balance model complexity with expressivity. I would imagine others will try to build on this work and use it in their applications. ** Overall I think this is a good quality submission, with interesting theoretical results. However, I strongly suggest the authors include the suggested additional experiments. ** Author Response Thank you for your response. I am glad to hear you are experimenting with other datasets and metrics: the inclusion of these will make the empirical evaluation of your model more convincing. Reviewer #4 made a good suggestion to compare to truncated identity and I hope this comparison will also be included in the final paper, as well as a discussion as to why using the truncated identity matrix of rank k can be seen to be less expressive than other rank k matrices one could use for Cd (for example, by comparing rank 1 truncated identity with HolE). [1] Tim Dettmers, Pasquale Minervini, Pontus Stenetorp, Sebastian Riedel: Convolutional 2D Knowledge Graph Embeddings. AAAI 2018 [2] Timothée Lacroix, Nicolas Usunier, Guillaume Obozinski: Canonical Tensor Decomposition for Knowledge Base Completion. ICML 2018

Reviewer 2



This paper proposes HolEx, a method that uses a combination of purturbed Holographic Embeddings (HolE) for knowledge graph completion. STRENGTH - Nice presentation - HolEx achieves a performance that surpasses the state-of-the-art in the FB15K knowledge graph completion task. - Extensive theoretical analysis of the proposed method and its performance. WEAKNESS - Some might say the approach (ensemble) may not be novel enough. - Empirical evaluation on other datasets may be desirable, such as WN18(RR) and NELL995. - Connection to complex embeddings may be desirable, which is known to be isomorphic to HolE. COMMENTS This is a solid piece of work that extends HolE, in somewhat straightforward manner in terms of approach, nonetheless it is quite effective. Moreover, theoretical analysis provided is very nice. I would like to see the discussion on its relationship to ComplEx, which has recently been shown to be isomorphic to HolE [Hayashi and Shimbo, ACL 2017], on which HolEx is based --- this is because ComplEx can be computed in linear time while HolE is not. Some minor suggestions: Since there is space on the left and right side of Table 1, please report Mean Reciprocal Rank and HITS@1, 5 as well, for convenience of comparison with existing work that only reports these numbers. Regarding Remark 1, could you clarify the situation of the numbers for ProjE in Table 1? I checked the GitHub page for ProjE but could not find the numbers on the table there (although there is a discussion in Issue tracker but no numbers are present.)

Reviewer 3



In this paper, the authors introduce a method to interpolate between RESCAL and HolE in the setting of knowledge base completion. Whereas RESCAL models the score of a triple (o, r, s) as (with x the tensor product), HolE models the score of a triple (o, r, s) as where conv is a circular convolution. The authors show that by adding a hadamard product with e_o and a vector c_j, with k independent c_j, we obtain a rank-k linear transform of the tensor product e_o x x_s. Thus, by increasing the rank, the authors interpolate in expressivity between RESCAL and HolE. The next part of the paper introduces 2 specific bases, based on haar wavelets and bernoulli vectors. Finally the authors test their algorithms on FB15K. I like the main idea to interpolate between HolE and RESCAL, but didn't understand section 3.1 and 3.2. It seems to me that the only criteria for expressivity is the rank of C_d. In this case, why not use a truncated Identity matrix for C_d? Why is there a difference in performance between the Haar basis and the bernoulli basis ? Regarding the Haar basis, why are we happy to get partial sums of the diagonals ? As currently explained, I didn't understand the value of theorem 2 and 3. Why would we want to "pin-point" (and what does it mean) non-zero values? is it realistic to have a sparse tensor product matrix ? (embeddings are not asked to be sparse, I don't see where the sparsity in this matrix would come from) I think this part should be re-written, maybe using figures, and theorem 2/3 should be written in a more formal way. The same comment on formality applies to the proofs of theorem 1 and 2, which are elementary but hard to follow because objects are not properly defined before they are used. It seems natural to introduce what would happen with a truncated identity matrix. This would amount to only using the k first values in the left-hand side vector, which would lead to a RESCAL with a matrix of size (k,d) pre-multiplied by a projector on the first k vectors. This begs the question, is this method very different from a low-rank RESCAL ? I think this comparison would be much more informative than the current section 3. Regarding experiments, I find the state of the art badly represented. Notably "Knowledge Base Completion: Baselines Strike Back" by Rudolf Kadlec et al. obtains a hit@10 of 89.3 with DistMult, which is higher than the results reported here for HolEx. Thus, the claim in the abstract about beating the state of the art seems wrong. ---------------------------- AFTER REBUTTAL ---------------------------- Thank you for your feedback, and for discussing the truncated identity matrix solution. Since this doesn't work as well as the Haar or Bernoulli "basis", I would think the real justification for what this work does lies in fourier space, rather than in a low-rank approximation of the tensor product. After discussion, I change my decision to a 6. I believe that the theory is not currently satisfying and / or supported by experiments. - low rank approximation of tensor product doesn't seem to be what makes this work. - no experimental evidence that the tensor product estimated are sparse. (which kills justification based on sensing) (regarding the author's feedback, maybe the type "person" won't be encoded on one dimension, but on a linear combination of dimensions. Since everything is initialized at random, I find it hard to believe that the canonical basis will have any meaning in the final embeddings) - no experimental evidence of a full "interpolation" between HolE and RESCAL (i.e., an experiment that shows HolE and RESCAL performances and how this method's performance get close to either HolE or RESCAL depending on the size of the basis used) With these, this paper would easily go from a 6 to a 9.